# Acceptability of minimally invasive autopsy by community members and healthcare workers in Siaya and Kisumu counties, western Kenya, 2017–2018

**Peter Otieno**[1]*, **Victor Akelo**[2], **Sammy Khagayi**[1], **Richard Omore**[1], **Kelvin Akoth**[1], **Maryanne Nyanjom**[1], **Sara Ngere**[1], **Ken Ochola**[1], **Maria Maixenchs**[3], **Ahoua Kone**[4], **John Blevins**[4], **Emily Zielinski-Gutierrez**[2], **Beth A. Tippett Barr**[2,5]*

1 Kenya Medical Research Institute, Center for Global Health Research, Kisumu, Kenya, 2 U.S. Centers for Disease Control and Prevention, Kisumu, Kenya, 3 IS Global, Hospital Clínic-Universitat de Barcelona, Barcelona, Spain, 4 Emory University, Atlanta, Georgia, United States of America, 5 Nyanja Health Research Institute, Salima, Malawi

* otienoop2017@gmail.com (PO); Beth@nyanja-health.com (BATB)

**Data Availability Statement:** Qualitative data has been made available at https://data.mendeley.com/datasets/57mwc7kwds/1.

## Abstract

Worldwide, nearly six million children under the age of five (<5s) die annually, a substantial proportion of which are due to preventable and treatable diseases. Efforts to reduce child mortality indicators in the most affected regions are often undermined by a lack of accurate cause of death data. To generate timely and more accurate causes of death data for <5s, the Child Health and Mortality Prevention Surveillance (CHAMPS) Network established mortality surveillance in multiple countries using Minimally Invasive Tissue Sampling (MITS) in <5 deaths. Here we present acceptability of MITS by community members and healthcare workers in Siaya and Kisumu counties, western Kenya. From April 2017 to February 2018, we conducted 40 in-depth interviews and five focus group discussions with healthcare workers and community members, before and during CHAMPS implementation. Participants were purposively selected. Field observations to understand traditional death-related practices were also performed. Interviews were transcribed into Nvivo 11.0 for data organization and management. Analysis was guided by the grounded theory approach. Facilitators of acceptability were desire to understand why death occurred, timely performance of MITS procedures, potential for MITS results in improving clinical practice and specific assistance provided to families by the CHAMPS program. However, cultural and religious beliefs highlighted important challenges to acceptability, including CHAMPS teams recruiting after a child's death, rumours and myths, unmet expectations from families, and fear by healthcare workers that some families could use MITS results to sue for negligence. Increasing MITS uptake requires sustained strategies to strengthen the identified facilitators of acceptability and simultaneously address the barriers. MITS acceptance will contribute to better characterization of causes of death and support the development of improved interventions aimed at reducing <5 mortality.

**Funding:** This study was funded by Bill and Melinda Gates Foundation as part of the Child Health and Mortality Prevention Surveillance (CHAMPS) program (grant no. OPP1126780). The funders had no role in study design, data collection and analysis, decision to publish or preparation of the manuscript.

**Competing interests:** The authors have declared that no competing interests exist.

## Introduction

Although global progress in reducing the mortality rate among children under the age of five years (<5s) has been considerable, disparities exist between regions and countries [1]. Worldwide, nearly six million <5s die annually. A substantial proportion of these <5 deaths can be attributed to diseases that are potentially preventable and treatable through simple, affordable interventions [2]. The highest burden of <5 mortality is in sub-Saharan Africa with 1 in 13 dying before their fifth birthday, a rate 14 times higher than the average observed across high income countries [1]. The causes of death differ substantially from one country to another, highlighting the need to broaden understanding of child health epidemiology at a country level rather than in large geographic regions [3]. Within countries such as Kenya, the burden of under-5 mortality differs further by region, urban and socio-economic status. For example, a child born in the western Kenya area formerly known as Nyanza province is almost twice as likely to die before age 5 than a child born in the Central region [4]. The Nyanza region is made up of six counties namely Siaya, Kisumu, Homabay, Migori, Kisii and Nyamira. Recent data for Kisumu and Siaya counties-the counties in which this study was conducted- estimate an <5 mortality rate of 79 and 81/1000, respectively [5–7].

Efforts to improve child mortality indicators in the most affected regions are often undermined by a lack of accurate and complete cause of death data [8]. Available sources such as vital statistics registries and hospital records are characterized by incomplete and simplified information that underestimate national and global disease burdens, consequently limiting the design and implementation of effective intervention strategies [9]. Where available, the cause of death reported is often a single, non-specific cause that doesn't allow for better understanding of the complex factors contributing to child deaths [10].

Conventional autopsy can provide more definitive cause of death data. However, this is invasive, requires highly skilled training, strong lab capacity [11], and potentially violates cultural respect for the dead [12]. The logistical and financial challenges associated with autopsy generally exceed the capacity of families and government health systems, and acceptability of post-mortem approaches by communities and healthcare workers is not well understood, especially where stillbirths and young children are involved.

In order to generate timely and more accurate causes of death data for <5s, the Child Health and Mortality Prevention Surveillance (CHAMPS) Network, supported by the Bill & Melinda Gates Foundation, has established mortality surveillance in multiple countries using Minimally Invasive Tissue Sampling (MITS) [13]. MITS has emerged as a preferred alternative to more invasive procedures as it has comparable accuracy in determining cause of death, particularly for infectious conditions [14]. In the conduct of MITS, small samples are collected from key organs using biopsy needles and examined in the laboratory through multiple testing methods. This less invasive approach only targets certain organs rather than opening the body as done in complete diagnostic autopsy [15].

MITS procedures offer the possibility for gathering critical missing data to determine the causes of under-five mortality; however, because the procedure is carried out on the body of a recently deceased child, and in this case, in a setting where autopsy is not commonly practiced, a wide range of complex religious, cultural, and ethical questions inevitably arise. Widespread acceptability and success of child mortality surveillance incorporating MITS requires a profound understanding of death related cultural and religious norms and practices. For example, beliefs about death and the afterlife, opposition to and concerns about body disfigurement, difficulties in obtaining consent from grieving families, inadequate involvement/endorsement of community leaders, lack of community awareness, suspicion of researchers, and burial practices are some of the factors underlying autopsy refusal [16]. When introducing a new and

potentially controversial procedure such as MITS, a better understanding of these kinds of cultural norms and practices is essential not only for acceptability but also for the long-term sustainability of child mortality surveillance.

The acceptability of MITS to recently bereaved parents and community members in Western Kenya was not well known prior to study implementation. Previous studies have assessed hypothetical acceptability of MITS and have suggested the need for further validation during real-life MITS implementation [17, 18]. Here we present a qualitative assessment of the specific cultural, religious and socio-behavioural factors that influenced acceptability of post-mortem diagnostic procedures on stillbirths and deceased children <5s. These findings were generated as part of a formative research study to examine the acceptability of implementing mortality surveillance using the MITS procedure prior to the launch of such surveillance activities in 2018 as well as interviews conducted after MITS was launched. As such, some responses reflect hypothetical perspectives about MITS acceptability whereas others reflect perceptions after the procedure had been implemented. These differences are explored in the discussion section of this paper.

## Methods

### CHAMPS overview

The CHAMPS Network was established in 2016 to collect robust, timely and accurate longitudinal mortality data in a network of sites with the overall objective to better understand and track preventable causes of childhood death in areas that experience high child mortality rates [19]. CHAMPS data collection includes clinical data abstraction, verbal autopsy and MITS. The biological specimens collected by MITS procedure are evaluated for evidence of infectious or other diseases locally and at the CHAMPS Central Pathology Laboratory in Atlanta, USA. All information is then compiled into a single file for each child and a panel of medical experts and epidemiologists meet routinely to review files, agree upon the immediate, underlying and comorbid causes of death, and assign relevant codes from the International Classification of Diseases, Tenth Revision (ICD 10). Details on how causes of death are determined has been published elsewhere [20]. The results on the causes of death obtained, are shared among national health institutions, communities and families. The aggregate CoD findings also inform broader public health action to reduce <5 mortality in the nine countries where CHAMPS operates. In Kenya, CHAMPS activities are implemented in two different Health and Demographic Surveillance (HDSS) catchment areas (Karemo in Siaya and Manyatta in Kisumu) in order to maximize the notifications of community-based deaths and understand <5 deaths in the context of population level health [6, 21, 22]. In the initial stages, the site had designated only one MITS performing centre-Jaramogi Oginga Odinga Teaching and Referral Hospital in Kisumu. This meant that MITS cases enrolled at the rural site in Siaya had to be transported by road to Kisumu for MITS performance, and then back for burial. The first MITS in Kisumu was performed in May 2017 and in Siaya in August 2017. At the end of the data collection, a total of 70 MITS had been performed. The establishment and operations of the HDSS have been previously described [21].

The CHAMPS' socio-behavioural science team engaged community members and leaders to sensitise them on child mortality surveillance activities, to understand their perceptions of child death and MITS, and to monitor and address emerging rumours in the communities where CHAMPS is implemented.

### Study design

This qualitative study in the formative research phase was guided by a descriptive phenomenology and grounded theory approaches. Grounded theory is a method which explores a topic

without relying on any pre-determined constructs to establish a frame of inquiry in order to minimize researcher bias [23]. Phenomenology is a qualitative research method in which respondents' views and perceptions are explored as they were experienced or perceived in real life. This study explored perceptions about the acceptability of the MITS procedures using a phenomenological method that pre-determined certain topics relevant to informing how MITS could be conducted and a grounded theory method to understand religious and cultural beliefs, practices, and rituals related to the phenomenon of death without a pre-determined focus on topics.

## Data collection

Participants were identified using purposive sampling technique. The categories of participants included administration officials, village elders, religious leaders, traditional birth attendants, community health volunteers, healthcare providers and parents/caregivers who had previous experience with child loss.

From April 2017 to February 2018, we conducted 40 in-depth interviews comprising 29 key informant interviews (KII) and 11 in-depth interviews (IDI) using a semi-structured interview method. We also conducted a total of five focus group discussions (FGDs) with 8 to 10 participants in each FGD. Two FGDs were conducted in peri-urban Kisumu and three in rural Siaya, with 48 total participants. Prior to conducting these activities, we developed one interview guide for key informants, a related guide for general in-depth interviews, and a guide for conducting FGDs. The semi-structured format assured that key topics were discussed while allowing the interviewer to probe on specific information provided in any given interview. These interviews were conducted as part of the formative research phase before mortality surveillance was launched in Siaya and Kisumu and as such surveillance was implemented. KIIs included open-ended verbal questions with targeted respondents who were deemed to have first-hand knowledge of their community's history, norms and values [24] including persons who had experiences in proceedings related to death and dying. IDIs were carried out to collect views, concerns/experiences and expectations of individuals representing entities or institutions that were to participate in CHAMPS activities. The FGDs were carried out for overall understanding of views among groups, especially in instances when it was deemed that a more productive and robust conversation could occur due to group dynamics. FGD participants were drawn from a cross-section of general community members representing various areas of interest to the study such as age, gender, religion, education level, experience of child loss and participation in CHAMPS. Interviews with community members and community health volunteers were conducted in Dholuo (the local language) or in Kiswahili (national language), and most professional health care providers chose to be interviewed in English. Informed consent was obtained from all participants before interview. Participants were informed about the study, its purpose, potential risks and benefits, their rights, and how their information would be used. Permission for audio-recording of interviews was sought, including explanation on storage and use of the recordings. For the FGDs, individual consent was obtained from each participant and the need for confidentiality within the group was emphasized.

Finally, we conducted 7 field observations for specific events including the initial visit to families after child death, the process for obtaining consent from families, and the process for removal of the body from the morgue after the MITS was completed so that it could be prepared for burial. Study staff accompanied families on three different occasions as bodies of their children were removed from the morgue after MITS procedures and transported to burial sites. This multi-method approach was aimed at facilitating data triangulation to help validate information collected by different methods regarding the acceptability and perceptions of CHAMPS activities.

## Analysis

Interviews and the focus group discussions were audio recorded and later transcribed and translated into English before uploading into Nvivo 11 software for coding and analysis [25]. Simultaneous transcription and translation was undertaken by two trained research assistants with multilingual proficiency. The transcriptions were then reviewed by the local research supervisor to ensure accuracy of the translation, representation of the participants' messages and maintain the original intent, tone, and context.

After separately reading and re-reading the transcripts and field notes to gain a comprehensive understanding of the content and context of the data, the research team sat together to develop a coding framework. We created a codebook that outlined the categories or themes we wanted to explore in the data. codebook provided a structure for organizing and analysing the data. By way of open coding, we broke down the data into smaller segments and assigned initial codes to them by highlighting sections of text that related to specific concepts, ideas, or themes., organize the codes into a coding system. Later, we grouped related codes together under broader themes. This helped to create a hierarchical structure that reflected the relationships between codes.

Data were analysed using an inductive approach [26].Data from field observations were collected in brief field notes and analysed thematically.

The conceptual framework for this qualitative assessment was anchored broadly on acceptability as a construct. We defined acceptability as the level of willingness/desire of an individual and/or community to allow MITS to be conducted on a deceased child based on social, cultural and religious influences. The in-depth interview guides for key informants and focus group discussions as well as observation guides were developed using the conceptual framework shown in Fig 1.

a. Beliefs about child death and corpse, religions and traditions, confidentiality, family issues, perceived need and appropriateness, etc.

b. Desire/willingness to consent and gain knowledge of the cause of death

c. Rituals and grieving

d. Perceptions about specific assistance offered by the CHAMS Kenya program towards funeral costs

e. Beliefs about early pregnancy loss, still birth and neonatal death (i.e. religious and traditional beliefs, confidentiality, family issues, appropriateness of MITS)

f. Perceptions of the individual, community, and social benefits of child mortality surveillance

g. Community perceptions about the capacity and quality of ANC and delivery

h. Relevant cultural practices

i. Stigma associated with stillbirths and neonatal deaths

j. Community understanding and acceptance of public health initiatives such as CHAMPS including a general history of public health interventions in the target communities

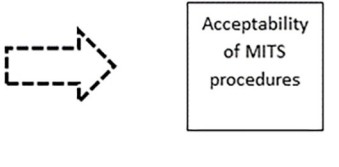

Acceptability of MITS procedures

**Fig 1. Descriptive illustration of variables explored as possible influencers of MITS acceptability.**

### Ethical approvals

Ethical approval for this study was provided by the Kenya Medical Research Institute (KEMRI SSC protocol # 3313) and the US Centers for Disease Control and Prevention (CDC protocol # 6916) ethical review committees.

## Results

Fourteen KII participants (48%) had experienced death of a child under five while 10 (35%) were healthcare providers based at different health facilities within the CHAMPS surveillance area. None of the healthcare workers interviewed had observed the MITS procedure. Other participants included general community members and leaders. Socio-demographic characteristics of the FGD, KII and IDI participants are presented in Table 1.

### Knowledge and perception of post-mortem examination by community members

Previous experience and existing perceptions about post-mortem examinations by community members was varied among participants. A commonly held view was that post-mortem examination was only meant for police and judicial investigations in which there is suspicion of wrongdoing, or something only families of higher socioeconomic standing could afford. Most reported that they had no prior experience with post-mortem examinations, particularly in stillbirths and young children. This unfamiliarity meant that participants were not aware of the details involved in any post-mortem procedure. Some of the respondents who had heard about the MITS procedure through community engagement activities that occurred prior to commencing surveillance reported their understanding that MITS would not require taking samples from the body; rather, they believed that it involved a doctor examining the dead body (without performing any procedure) and coming up with a report on cause of death. Such respondents also wondered why cause-of-death results could not be available the same day as the procedures.

> "*During the post-mortem, the pathologist asked which part of the body he was complaining of and we said chest, when he looked at the chest he said it was pneumonia and he did not remove any part. The second one was an accident with head injuries and when they examined the head, they found blood clots and said there was bleeding which went to the brain and the pathologist gave response there. I did not see any sample being carried to the lab; just written there and everything was closed*" (FGD, Manyatta)

Some participants explained that they saw little value attached to post-mortem examination and learning the cause of death because of the irreversibility of death. There was also a belief that death was a destiny which cannot be prevented:

> "*Now if we did a post-mortem, what will it help us with . . ..in Manyatta as a whole I don't think if people take post-mortem seriously. To my knowledge when someone dies, he is [already] dead. They will ask you, if you do a post-mortem is there a reversal to the death*?" (Male participant, FGD Manyatta)

### Barriers to acceptability of MITS

Although there was growing acceptability of MITS among community members and healthcare workers, there were several factors that potentially affected how MITS was perceived and

**Table 1. Background characteristics of the participants (n = 88).**

| Participant characteristics by Interview type | | Number of participants | | |
|---|---|---|---|---|
| | | Key Informant interviews (KIIs, n = 29) N(%) | In-depth interviews (IDIs, n = 11) N(%) | Focus group discussions (FGDs, n = 48 participants) N(%) |
| Gender | Male | 15 (52) | 4 (36) | 18 (38) |
| | Female | 14 (48) | 7 (64) | 30 (62) |
| Age group | <30 years | 5 (17) | 2 (18) | 23 (48) |
| | 31–49 years | 9 (31) | 8 (73) | 17 (35) |
| | 50+ years | 15(52) | 1 (9) | 8 (17) |
| Education | None | 0 (0) | 0 (0) | 1 (2) |
| | Primary | 14 (48) | 0 (0) | 20 (42) |
| | Secondary | 7 (24) | 1 (9) | 19 (40) |
| | Tertiary | 8 (28) | 10 (91) | 8 (16) |
| Site/Area | Manyatta (Kisumu) | 16 (55) | 6 (54) | 22 (46) |
| | Siaya | 13 (45) | 5 (46) | 26 (54) |
| Primary care giver of child <5 years | Yes | 28 (97) | - | 36 (75) |
| | No | 1 (3) | - | 12 (25) |
| | N/A | 0 (0) | 11(100) | |
| Had a child enrolled in CHAMPS | Yes | 0(0) | - | 1 (2) |
| | No | 0(0) | - | 31(65) |
| | N/A | 29(100) | 11 (100) | 16 (33) |
| Religion | Christian | 25 (86) | 10 (91) | 45 (94) |
| | Muslim | 2 (7) | 0 | 2 (4) |
| | Other | 2 (7) | 1 (9) | 1 (2) |
| Marital Status | Single | 3 (10) | 2 (18) | 6 (13) |
| | Married_ monogamous | 19 (66) | 8 (73) | 34(71) |
| | Married_ polygamous | 4 (14) | 0(0) | 4 (8) |
| | Divorced | 0(0) | 0(0) | 3 (6) |
| | Widowed | 3 (10) | 1 (9) | 1 (2) |
| Main Occupation | Formal employment | 10 (35) | 9(82) | 1(2) |
| | Informal employment | 3 (10) | 1 (9) | 5 (11) |
| | Self-employed | 15 (52) | 1(9) | 27 (56) |
| | Other | 1 (3) | 0 | 15 (31) |
| Ever lost a chid <5 years | Yes | 14 (48) | - | 31 (65) |
| | No | 15 (52) | - | 17 (35) |

0 (Zero) denotes "None"

- (Dash) denotes that question on the attribute/characteristic was deemed Not Applicable to the interviewee

agreed to, including timing of CHAMPS involvement after death, concerns by healthcare workers, cultural and religious beliefs, rumours and myths, general unfamiliarity with post-mortem procedures and unmet expectations from bereaved families.

## Timing of CHAMPS involvement

Participants expressed a strong desire for CHAMPS to assist caregivers in accessing quality healthcare before a child died and not only to "show up" when death occurred.

"*Why do you show up only after the child is dead? When my child got sick, I rushed to the hospital, but I was turned away because health workers were on strike. I did not have the money to go to a private clinic, so I just bought some pain killer for the child. . . and came back home.*" (Female respondent, KII, Manyatta)

## Healthcare worker concerns

Healthcare workers expressed concerns that MITS results could be used to evaluate their competencies especially if the results differed from clinical management decisions made during care. They also feared that some families could use the results to sue for negligence:

"*What will community members think about us or our health facilities if MITS results show that a child was being managed for a different diagnosis than the determined cause of death?*" (Healthcare worker, IDI, Kisumu)

## Cultural and religious beliefs

The notion that results from MITS would be used to prevent future <5 child deaths was not easily understood by some respondents who believed that human medical intervention could not prevent death because it was God's plan. There was also the perception that post-mortem examination was not necessary as it would not bring the deceased child back to life:

"*Let the child rest. It was God's will to take the child*" (Religious Leader, KII, Siaya)

"*Children are innocent pure souls whose dead bodies should not be taken through the disturbance that is post-mortem*" (Mother, FGD, Kisumu)

Respondents also shared that beliefs related to timing of burial were a potential impediment to acceptability. It is a common cultural practice among the Luo, a predominant cultural group in western Kenya, to bury young children soon after death, especially in the event of death occurring at home. There is also a prevailing belief that for twins, the death of one could adversely affect the surviving twin. If one died, the burial was conducted even more quickly than would be generally practiced with children:

"*For twins, if one of them is dead and is being buried, the other should not be nearby. He is usually hidden and must not be exposed to the burial process. If he gets to know what is happening, he may follow the dead one. Also, we do not want a situation where the mother is crying that one child has died while the twin brother is still alive. We do not want her to cry because if she cries then she breastfeeds the live one, this can affect it.*" (Male participant, FGD, Siaya)

In other instances, it was explained that the need to bury quickly was motivated by a perceived need to heal and "get over" the memories of the painful loss. It was uncommon to transfer bodies of deceased children to the morgue for preservation. For the period in which later interviews and focus groups were held, MITS procedures for both community and health facility deaths were conducted at a morgue within the primary referral facility. It was considered taboo for the body to leave the compound if a grave had already been dug, which led to families rejecting participation in MITS in at least one occasion. Members of some religious faiths and local sects in both Siaya and Manyatta-Kisumu did not prioritize scientific or conventional medicine and did not easily understand the importance of post-mortem examination.

## Rumours and myths

Rumours and myths about MITS procedures were also cited as a barrier to acceptability. One common rumour was that the procedure involved harvesting of body organs for some cultic or commercial purposes, e.g. selling of organs.

"*. . .there are some thoughts you hear people saying that when my person is operated there are some things that they remove from him/her—at times, someone's heart is removed and taken away*" (FGD, Siaya).

Some reported hearing that the CHAMPS program was paying participating families in exchange for the bodies of children to be enrolled for MITS. These rumours were not directly associated with families who had consented for MITS but originated from their neighbours or persons who had not had any direct experience with the MITS procedure.

## Unfamiliarity with post-mortem examination and medical diagnosis

Some families who had no prior experience personally or in the community with post-mortem examination for children or otherwise had difficulty consenting to MITS:

"*We have never seen any child in this community taken for post-mortem, our child cannot be the first one. . .*" (Family in Siaya; informal conversation during field observation)

Some of the families whose children had undergone MITS were skeptical about the results where it revealed an unfamiliar condition. For instance, a couple whose child died of sickle cell disease doubted MITS results because they had not heard about the disease before.

"*I still have doubts about the death. . . about the results I received. Even my wife had not heard about sickle cell. We had not heard about sickle cell and I did not accept that my child died of sickle cell. But I told her that they have brought the results and is what we received, but I could see that she also had doubts*" (Male respondent, KII, Siaya)

## Unmet expectations from families

There were also cases on unmet expectations from some of the families whose children had been enrolled for the MITS procedure. For instance, some families expected CHAMPS program to meet the cost of mother's treatment for illness or condition linked to child's death:

"*My expectation was treatment to my body because I may be having an infection since even the placenta was rotten*" (Mother who experienced a stillbirth, Field Observation during MITS results delivery, Kisumu).

In another case, the cause of death results showed that a mother had passed bacteria to her unborn baby leading to the baby's death from sepsis. The CHAMPS Kenya clinicians who delivered the result advised her on a possible course of treatment for her but she reported in the focus group that she had no money to meet the treatment costs. During an industrial strike by public healthcare workers that took place during the study period over a pay dispute, respondents expressed concern that the CHAMPS Kenya program did not assist them to access healthcare for their sick children.

### Facilitators of MITS acceptability

Facilitators for the acceptability of MITS included desire to understand cause of death, timely performance of procedures, potential for MITS results to improve clinical practice and assistance provided to bereaved families by the CHAMPS program.

### Understanding why death occurred

The desire of parents to understand their child's death was one driver of MITS´ acceptability. Even if some community members did not completely agree with the practicalities related to performing the MITS procedure, such as transferring the bodies over a distance to the morgue in Kisumu, they still reported that the objective to know a child's cause of death was important. This was particularly evident among participants who knew someone in their close social circle or family who had experienced multiple pregnancy losses or child deaths:

> "*I consider it very important to know what has caused the death of a child so that we can prevent similar deaths if it is a condition that is preventable*" (Male respondent, KII, Siaya)

> "*They told me that the reason for carrying out [MITS] on the child was to know the cause of death and that the knowledge could be of help to me or community in taking care of other young children so that they will not lose their lives. That's what touched my heart and I decided to agree so that the truth is known*" (Mother who had lost a child <5 years, KII, Siaya)

Participants also reported instances where community members would attribute the death of a child to cultural beliefs such as witchcraft or claims of parental negligence. This resulted in stigmatization of mothers for allowing or indirectly causing the death. In such instances, MITS results provided relief and de-stigmatized the parent:

> "*People said that my child had the big disease-HIV, and that I am the one who knew where the disease was from. So, when the results were delivered and I was told that it was pneumonia, I found some relief* ". (Female respondent, KII, Siaya)

### Timely performance of MITS procedures

Based on findings from these interviews, the CHAMPS team expedited the process for conducting MITS so that families received the bodies of their children in time to prevent delay of burial. This emerged as a key factor for MITS acceptability. Respondents emphasized that MITS procedures should not interfere with families' desire to bury their child in a timely manner.

> "*As Muslims when a person dies or a child it's an easy task. We only deal with burial plans. Most of the time burial plans are on our minds. The only thing that we do to the corpse is to wash it. After washing it and wrapping it well that is it. We pray for him and bury him the same day.*" (Muslim Religious Leader, KII, Siaya).

> "*When a small child dies they should be buried by noon for example the one who has just died if they are months old or are a year old they can prepare and have the burial by 10am or 11am. Even if people are staying behind in the funeral but the child has already been buried. . . . these are things I have experienced. I lost a child who was months old, and he was buried before 12 noon. I have also buried two other children who died immediately after birth . . .*

*they are buried immediately. They do not stay for long"*. (Mother who had lost a baby, KII, Siaya)

## Potential for MITS results in improving clinical practice

Healthcare workers saw value in knowing cause of death and believed it could improve their practice when MITS results are shared and discussed during existing mortality review meetings conducted at health facilities:

"*For mortality statistics, I think it informs our clinical practice. . . It helps us know how to take care of the ones who are alive. If the commonest cause of death for us is this, or these are our top 10 then it informs our medical practice to see if we need to do something specific to intervene on those common causes".* (Healthcare worker—County Referral Hospital, KII, Siaya)

## Assistance provided to families by the program

Specific assistance was provided by CHAMPS Kenya program to appreciate and/or compensate participating families for their time. From the perspective of the families, this was an important incentive to participate. In keeping with the traditional practice in this part of Kenya of making monetary or material contributions to bereaved families towards funeral expenses, CHAMPS Kenya makes a modest contribution of 5,000 Kenya shillings (approx. USD 50) to parents of deceased children who participate in surveillance. After MITS is conducted at the morgue, the body is also transported free of charge to the place of burial. For families that live within the urban HDSS in Kisumu, the preferred burial place is their rural homes outside town. Parents reported that it is helpful and culturally acceptable to families when the study provides this transportation expense.

"*When my child died. . .. they told us that if the child was to be buried at home they would cater for the transport of the child and parents to the rural home and a little money which we used to prepare tea for guests*" (Male parent, FGD, Manyatta).

## Discussion

This study highlights facilitators and barriers of MITS in children as perceived by community members and healthcare workers in western Kenya. Our findings indicate that facilitators central to acceptability of MITS included: parental desire to know the child's cause of death, especially in situations where perceived cause of death could result in stigmatization; specific assistance towards funeral costs provided to families by CHAMPS Kenya program; and the ability of the program to perform MITS procedures in a timely manner, returning the bodies to families without causing delay to burial plans. These findings are consistent with previous studies [27, 28]. It has also been previously reported that the potential for MITS to provide a learning opportunity for improved clinical care practice was a key facilitator for acceptability among healthcare workers [29, 30].

Even where there was hypothetical acceptance of MITS among general community members, the continued positive perception of MITS once the procedure had been implemented by bereaved families remains crucial to the success of mortality surveillance that uses MITS. The positive perception would affect how family members describe the process to other

community members thereby enhancing the program's reputation and credibility. Further, the continued positive perception would ensure that participating families look up to, and engage freely with subsequent activities such as results delivery and community feedbacks. To achieve this, a detailed, interactive consenting process that simplifies details of the procedure is important. A previous study in Soweto, South Africa also observed that bereaved parents struggled to understand the medical nature of MITS procedure and the technical language used during the consenting process [31]. Whereas the desire to know the cause of death was an important facilitator of MITS acceptability, our study also found that the potential use of MITS results to prevent similar deaths in the future or to improve child survival seemed less convincing to some bereaved families. Another previous study in Zambia found that some of the parents who declined autopsy on their children did so on the grounds that it would be a "waste of time," as the diagnosis should have been made in life and the findings would now be of no benefit to them [32]. In our study, the inconsistency of this finding with the desire to know cause of death is perhaps a result of unfamiliarity with the science of post-mortem in this setting and its perceived value in child mortality surveillance. The inconsistency may also be a demonstration of how qualitative research can uncover diverse and sometimes conflicting views on a single topic.

This study found that religious and cultural beliefs can strongly influence attitudes towards autopsies. The need for quicker burial for children, and the fear that autopsy constitutes a form of disturbance or harm to the dead bodies were some of the beliefs and practices associated with such perceptions. Previous studies have reported that removing body tissues for testing may be sensitive issues and a delay in returning the body for burial might also cause concern [17, 33]. MITS implementation should remain sensitive to such concerns and work to address them.

In this Kenyan study some healthcare workers feared that MITS would serve to unduly scrutinize their clinical competencies or potentially set them up for litigations related to professional negligence. This finding is also consistent with previous studies [18, 29]. Healthcare workers need continuous reassurance that the intention is not to blame them if MITS results differ from the clinical management decisions that were made during care but to identify gaps in clinical diagnosis or management from a broader systems perspective. Understanding both barriers and facilitators to MITS acceptability among healthcare providers was important for identifying education and sensitization approaches for various members of the community and for informing different procedures for death notification within facilities and within the broader community.

Post-mortem autopsies or MITS, particularly in young children, were not routinely practised in this setting. This therefore made it difficult for bereaved families to take the first leap into a process that they knew nothing about. This observation supports the finding of a previous study that showed that familiarity with post-mortem procedures provided reassurance to bereaved families that their child's body would be respected [34]. The criticism from community members that CHAMPS only focused on childhood death and that the program should be putting efforts into providing health services to children when they were sick rather than only showing up after children had died could be addressed through systematic data-to-action campaigns and community feedback sessions [35].

Increasing acceptance of MITS needs sustained efforts to mitigate these fears and concerns by enhancing the appealing attributes of the program while working closely with community and healthcare workers to address their fears and concerns. Since timelines of burial are a key priority for community members, acceptability for MITS is likely to increase if the procedure and related activities are conducted in a manner that does not delay burial timeliness set by families. There is also need for a proactive community engagement and rumour surveillance

systems that can unearth and address other barriers related to community perceptions, beliefs and practices.

It is imperative that social science knowledge and qualitative research guide the implementation of child mortality surveillance that uses MITS as they contribute to better understanding of specific cultural, religious and socio-behavioural factors that may increase or decrease acceptability of the MITS procedure on children, in a given setting. The knowledge obtained is also important to inform steps taken to improve program interaction with the community. This study therefore contributes to determining the overall acceptability of MITS in the context of child mortality in western Kenya as well as to inform the nature and scope of community engagement activities aimed at behavior/belief modification or community messaging for religious, traditional, thought/opinion, and political leaders.

Additionally, the study observed the contexts in which families grieve in the event of a child loss, to determine the best time and which family member(s) should be approached when introducing the MITS procedure. Our findings also demonstrate that exploring whether and in what context incentives, such as specific assistance with funeral costs or body transportation, would be ethically feasible, effective or appropriate would help to strengthen ongoing efforts to increase MITS acceptance. Increased MITS acceptance will likely enhance the use of postmortem examination and accurate cause of death determination as a key driving strategy for reducing <5 mortality.

We recognize that our study had some limitations. The majority of respondents interviewed were general community members who had not participated in MITS either as a parent of an enrolled MITS case, nor as a household or family member. They had no real life experience with the procedure. So any views expressed in favour of or against MITS were largely hypothetical and may not mirror the actual views and perceptions that may only become clear in the face of an actual MITS situation. Additionally, it is possible that the persons who agreed to participate in the interviews and focus groups were different and could have given different responses than those who did not participate. We tried to minimize this problem by recruiting well-informed community members and representatives, developing a large and purposive sampling frame and recruitment strategy, and assessing responses for saturation which demonstrated the predominant perspectives from the community had been heard.

Our findings provide important insights on areas that need strengthening in order to improve MITS acceptance and make it possible for the CHAMPS program and the Ministry of Health to better understand causes of child deaths in these settings and design more impactful interventions.

## Acknowledgments

We thank Dr Khatia Munguambe and Dr Elizabeth O'Mara for their support. We acknowledge the support of the County Department of Health for Siaya and Kisumu Counties (Kenya) as well as the communities/families in Manyatta (Kisumu) and Karemo (Siaya) for their participation and time. We particularly thank the Community Health Volunteers who assisted our data collection team with locating participants.

**CDC disclaimer:** The findings and conclusion in this report are those of the authors and do not necessarily represent the official position of the Centers for Disease Control and Prevention or the Agency for Toxic Substances and Disease Registry.

## Author Contributions

**Conceptualization:** Peter Otieno, Victor Akelo, John Blevins, Emily Zielinski-Gutierrez, Beth A. Tippett Barr.

**Data curation:** Peter Otieno, Maryanne Nyanjom, Ken Ochola.

**Formal analysis:** Peter Otieno, Kelvin Akoth, Maryanne Nyanjom, Sara Ngere, Ken Ochola.

**Funding acquisition:** Victor Akelo, John Blevins, Emily Zielinski-Gutierrez, Beth A. Tippett Barr.

**Investigation:** Peter Otieno, Kelvin Akoth, Maryanne Nyanjom.

**Methodology:** Peter Otieno, Kelvin Akoth, Ahoua Kone, Emily Zielinski-Gutierrez.

**Project administration:** Peter Otieno, John Blevins.

**Resources:** Victor Akelo, Richard Omore, Emily Zielinski-Gutierrez.

**Supervision:** Peter Otieno, Richard Omore, Emily Zielinski-Gutierrez.

**Validation:** Peter Otieno, Maria Maixenchs, Beth A. Tippett Barr.

**Visualization:** Peter Otieno.

**Writing – original draft:** Peter Otieno.

**Writing – review & editing:** Peter Otieno, Victor Akelo, Sammy Khagayi, Richard Omore, Sara Ngere, Maria Maixenchs, Ahoua Kone, Emily Zielinski-Gutierrez, Beth A. Tippett Barr.

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
