## [Decision Letter · Decision Letter 0]

20 Mar 2023

PGPH-D-22-01362

Acceptability of Minimally Invasive Autopsy by Community Members and Healthcare Workers in Siaya and Kisumu Counties, western Kenya, 2017-2018

Dear Dr. Otieno,

Thank you for submitting your manuscript to PLOS Global Public Health. After careful consideration, we feel that it has merit but does not fully meet PLOS Global Public Health’s publication criteria as it currently stands. Therefore, we invite you to submit a revised version of the manuscript that addresses the points raised during the review process.

Please see the reports from two reviewers below. Please consider carefully reviewer 2's comments about whether or not to split the work into two manuscripts. There are no word limits for this submission, so you are welcome to keep it as one manuscript, provided that you clarify in the way that the reviewer has suggested.

We look forward to receiving your revised manuscript.

Kind regards,

Hanna Landenmark

Staff Editor

Journal Requirements:

Additional Editor Comments (if provided):

Reviewers' comments:

Reviewer's Responses to Questions

**Comments to the Author**

1. Does this manuscript meet PLOS Global Public Health’s publication criteria? Is the manuscript technically sound, and do the data support the conclusions? The manuscript must describe methodologically and ethically rigorous research with conclusions that are appropriately drawn based on the data presented.

Reviewer #1: Yes

Reviewer #2: Yes

2. Has the statistical analysis been performed appropriately and rigorously?

Reviewer #1: Yes

Reviewer #2: N/A

3. Have the authors made all data underlying the findings in their manuscript fully available (please refer to the Data Availability Statement at the start of the manuscript PDF file)?

Reviewer #1: Yes

Reviewer #2: Yes

4. Is the manuscript presented in an intelligible fashion and written in standard English?

Reviewer #1: Yes

Reviewer #2: No

5. Review Comments to the Author

Reviewer #1: The manuscript is informative. I think please present the abstract section in structured format. please rewrite carefully the Introduction section with detail information and citied it with more references.

Reviewer #2: GENERAL COMMENTS

A general comment is that the community members and healthcare workers appear to be in opposite on their thoughts on the procedures. This would of course be similar to other studies on the same from other contexts (case in point being the Lawrence et al study which they have also referred to); however, the presentation of the views in this context is rather not enforcing a narrative but highlighting the differences and gives an impression of two papers (one on attitudes from community members and another on attitudes from healthcare workers) in one. It would be important that they are either merged as a narrative in which it is disclosed properly either in methodology or in the reporting of the findings OR the materials from healthcare workers is actually dropped in which case if it is enough it can be used for a different paper.

The language of the paper needs to be clear. In the introduction, it is indicated that “Here we present a qualitative assessment of the specific cultural, religious and socio-behavioural factors that influenced acceptability of post-mortem diagnostic procedures…” (lines 100 – 102), yet in reading the paper it shows that some of the participants had not had experience with the procedures and were asked in a rather hypothetical format. It would be important for the authors to highlight what really happened and what are they reporting on.

Also the language needs editing for coherence, for example there are references to post-mortem autopsy (e.g line 86), an autopsy is always post-mortem. There are also references of post-mortem examination, it would be helpful just to call it autopsy or choose one word which should be used uniform across. In lines 269 – 270 “Some of the families whose children had undergone MITS were reluctant to accept results where it revealed an unfamiliar condition…” I do not think this would be reluctance to accept results other than a denial of the information and the results. Reluctance to accept results gives an impression of them not showing up for the results. But they showed up and whether they liked it or not, they got the results. In processing the results, then there can be denial or scepticism. It is a matter of language which I think across the paper needs some sort of clarity and reinforcing.

On a general, the paper needs re-writing. There are important and interesting novel findings that it has but they are buried in a chunk of other ‘common’ findings of which some are similar to other studies from different contexts (e.g. Malawi and Mozambique). Other than focusing on the quality of the findings from the study, it feels rather that there was focus on the quantity of the findings and this is a disservice to an absolutely brilliant study which generated interesting and novel findings that has the potential of enriching the diversity of barriers as well as facilitators towards MITS and other health interventions reasons within the global health literature.

PARTICIPANT SELECTION (change the subheading: I would suggest data collection)

The first paragraph. The section is not clear. In one instance, it appears as if only Interviews (KIIs and SSI) were conducted yet again there is a mention of FGDs. Also, there is a need to highlight the actual categories of the people who were involved. They just refer to the people having some knowledge of the practices. Were they community leaders or just community members? This has been reported as a purposive sampling so there is a need to be very clear on the people who were purposively sampled.

Also, what is the difference between KIIs and SSIs? One can do a KII using a SSI, why are they treated as opposites in this reporting when the opposite of semi-structured is structured? In the lines 133-134 there are the words “KIIs included open-ended verbal questions…” and this is just a semi-structured approach (see, for example, Fylan, F., 2005. Semi-structured interviewing. A handbook of research methods for clinical and health psychology, 5(2), pp.65-78).

ANALYSIS

This study has been reported as having used a grounded theory analysis. However, it has also reported that they had a conceptual framework in place (which is sensible considering that there was a semi-structured and structured approach), it is therefore confusing how a grounded theory analysis was used.

How much influence did the framework have? How different was the data from the interviews to the recorded data from observations in as far as the study objectives are concerned that they had to have two different, and even contradictory, approaches for analysis? In the way that the results are presented and subsequently discussed, there is little evidence of a grounded theory analysis other than a thematic analysis. The authors should clarify on that within the analysis section which at this point is not clear.

BACKGROUND CHARACTERISTICS

Had a child enrolled in CHAMPS? Why is there the NA option? You can either have a child enrolled or not. That part on the table is not clear.

FINDINGS

KNOWLEDGE AND PERCEPTION OF POST-MORTEM

They should be separated. There is a conflation here of knowledge of autopsy and perceptions of MITS. The two are different yet that does not show in the presentation of this finding. It should be separated. Also, an engagement with the social characteristics of the respondents would provide significant and important clues. The section on the details of the participants is detailed but it does not show in the engagement with the analysis of the data. For example, as it is of qualitative research, one would expect that the categories of participant in lines 198 to 201 are identified with the aim of highlighting how his various social categories play into his response. It is not just his sex that would inform such a fatalistic position. Is he, for example, a religious leader? Do it as it has been in lines 298 – 299 where the context for the response is given. Further, the last part in lines 196 – 201, it reads like a point that belongs to the barrier aspect.

BARRIERS

TIMING

The heading might need to be changed just as the style of language of the paper. I take the paper as contributing to the larger body of MITS literature. The issue that is raised here, while applicable to a Kenyan experience, can however be interpreted within the broader dynamics of hospitals and politics within other settings. Broadening the conversation like that would be hugely relevant and strengthen the paper. I am thinking the barrier here would be factors beyond the MITS team, other than situating it in timing which for all practical purposes reads as if it was the timing of approach (that it is immediately after death; in which case it is discussed in the beliefs section) than that it is an issue of the political environment.

HEALTHCARE WORKER CONCERNS

Can the concerns be split properly as those of the public and healthcare workers? The two groups have different contexts in this research. I would recommend that the first part addresses the concerns from community members and then focus is on healthcare workers concerns which appear to be little than those of community members and there is no resolve of the same in the progress of the paper.

CULTURAL BELIEFS

229 – 231: The sentence needs rewriting. As it is, it gives an impression that it would be acceptable in adults as it can bring them back to life.

There needs to be an elaborate exploration of the time and periods of burial. As it is, it is rather vague and makes for difficult reading and appreciation of the context. It would be hard then to rely on such results in other contexts.

RUMOURS AND UNFAMILIARITY WITH AUTOPSY

These subheadings are almost communicating the same point. They are all related to the issues that have been discussed in Knowledge and perception (178 – 201). The issues discussed under the Rumours and Myths (255 – 262) can be merged with the issues on the perceptions towards MITS that are bunched together with the knowledge. The issues in the other subheading are a rehash of the arguments between 179 – 185, albeit in a different language.

The point in 269 – 276 is quite interesting that I think it should have been fully developed other than lumped together with the other repetitive point on knowledge of autopsy. It is actually a novel point from across other MITS studies and it would be relevant in the contribution of knowledge around MITS procedures.

UNMET EXPECTATIONS

The case in 285 – 288 is similar to the case made earlier in 269 – 276, yet these fall under two different subheadings and arguments. This again points to the need of harnessing the paper. These are also the interesting findings that would add to the breadth of literature on MITS and would even be applicable to other health interventions.

TIMELY PERFORMANCE

320 – 324: the quotation is not related to the argument advanced. Was the religious leader speaking as someone who had consented to MITS? The language in the sentences preceding the quote appear to indicate that this is a discussion on the reasons for consent to MITS, not what would be the probable reasons. In addition, the quotation coming after it is as well not communicating the same thing as the argument. The quotations are at best from people who did not consent for MITS yet the argument is for the reasons for consent to MITS.

DISCUSSION

Point between 370 to 372 is interesting as it contradicts studies in other contexts e.g. I would recommend that the authors engage with that literature as well which they have not done despite this being a discussion section. Also, a significant focus on such findings (which are not highlighted in the findings section as elaborately) would be important.

There is also an overlap between the discussion and recommendations such that the discussion is not properly developed. I recommend that the two areas are split in which the discussion is a section on its own and recommendations are in an own section. This will leave space for clear engagement with other literature as well as situating the findings within the body of relevant literature. As it is, little literature has been engaged with in the study and it shows within the discussion. Most importantly: where is the conclusion on the paper?

6. PLOS authors have the option to publish the peer review history of their article (what does this mean?). If published, this will include your full peer review and any attached files.

**Do you want your identity to be public for this peer review?** For information about this choice, including consent withdrawal, please see our Privacy Policy.

Reviewer #1: **Yes: **Dr. Tarana Jahan

Assistant Professor

Department of Medical Microbiology

Reviewer #2: No

---

## [Decision Letter · Decision Letter 1]

13 Jun 2023

PGPH-D-22-01362R1

Acceptability of Minimally Invasive Autopsy by Community Members and Healthcare Workers in Siaya and Kisumu Counties, western Kenya, 2017-2018

Dear Dr. Otieno,

Thank you for submitting your manuscript to PLOS Global Public Health. After careful consideration, we feel that it has merit but does not fully meet PLOS Global Public Health’s publication criteria as it currently stands. Therefore, we invite you to submit a revised version of the manuscript that addresses the points raised during the review process.

Thank you for revising your manuscript and for thoroughly addressing the comments of Reviewer 2. If you remember, the review by Reviewer 1 was rather short and inconclusive. Hence, a new (albeit, very positive) review has been provided by Reviewer 3 which focuses on improving the description of the qualitative methodology (particularly recruitment and more detail on the rigor of the analysis), and clarifying some points in the results and discussion. I am very sorry about the timing of second set of comments, but I think these clarifications will improve the readability and rigor of the manuscript greatly. Nonetheless, please let me know if you disagree with any of the requested changes, as you would with any reviewers' comments.

Good luck with your revision.

We look forward to receiving your revised manuscript.

Kind regards,

Ruwan Ratnayake

Academic Editor

Journal Requirements:

Additional Editor Comments (if provided):

Reviewers' comments:

Reviewer's Responses to Questions

**Comments to the Author**

1. If the authors have adequately addressed your comments raised in a previous round of review and you feel that this manuscript is now acceptable for publication, you may indicate that here to bypass the “Comments to the Author” section, enter your conflict of interest statement in the “Confidential to Editor” section, and submit your "Accept" recommendation.

Reviewer #2: All comments have been addressed

Reviewer #3: (No Response)

2. Does this manuscript meet PLOS Global Public Health’s publication criteria? Is the manuscript technically sound, and do the data support the conclusions? The manuscript must describe methodologically and ethically rigorous research with conclusions that are appropriately drawn based on the data presented.

Reviewer #2: Yes

Reviewer #3: Partly

3. Has the statistical analysis been performed appropriately and rigorously?

Reviewer #2: N/A

Reviewer #3: I don't know

4. Have the authors made all data underlying the findings in their manuscript fully available (please refer to the Data Availability Statement at the start of the manuscript PDF file)?

Reviewer #2: Yes

Reviewer #3: Yes

5. Is the manuscript presented in an intelligible fashion and written in standard English?

Reviewer #2: Yes

Reviewer #3: Yes

6. Review Comments to the Author

Reviewer #2: Thank you for addressing the comments and offering the rebuttals where necessary.

Reviewer #3: The paper aims to understand barriers and facilitators to the minimally invasive tissue sampling (MITS) for cause of death ascertainment used as part of a program whose goal is to reduce child mortality in high child mortality regions. It does this quite successfully, with a number of interesting and important findings in the work. The paper represents a solid contribution to understanding of autopsy procedures in regions where they may be needed most. Findings could potentially be generalized to research investigating mortality in other populations or using different post-mortem methods for cause of death ascertainment in the region. The methods were appropriate to address the research question and it appears that the study team reached saturation of the topic through its investigations.

Despite its strengths, there are several major gaps in the paper, which once addressed, will improve the quality of the paper, and make it more useful to the reader. To be publishable, authors should: address missing details in the introduction; provide a clearer description of methods, including who participated in the activities and why so many separate activities were undertaken; provide a clear description of what was done during analysis; describe the informed consent process; address inconsistencies in the results; describe how different groups and activities compared in the discussion; clarify or remove statements in the discussion that aren't found in the results; correct references.

The full line-by-line review has been uploaded as an attachment.

7. PLOS authors have the option to publish the peer review history of their article (what does this mean?). If published, this will include your full peer review and any attached files.

**Do you want your identity to be public for this peer review?** For information about this choice, including consent withdrawal, please see our Privacy Policy.

Reviewer #2: No

Reviewer #3: No

---

## [Editor Report · Decision Letter 2]

8 Aug 2023

Acceptability of Minimally Invasive Autopsy by Community Members and Healthcare Workers in Siaya and Kisumu Counties, western Kenya, 2017-2018

PGPH-D-22-01362R2

Dear Mr Otieno,

We are pleased to inform you that your manuscript 'Acceptability of Minimally Invasive Autopsy by Community Members and Healthcare Workers in Siaya and Kisumu Counties, western Kenya, 2017-2018' has been provisionally accepted for publication in PLOS Global Public Health.

Before your manuscript can be formally accepted you will need to complete some formatting changes, which you will receive in a follow up email. A member of our team will be in touch with a set of requests. In addition, the Editor has requested some minor changes below, which you can update at the same time.

Best regards,

Ruwan Ratnayake, LSHTM 

Academic Editor

Thank you for responding to another round of reviews so positively, and congratulations on being accepted! Please make the following minor updates to the text, which I trust can be done in the formatting stage:

- Replace "Atlanta, USA" with "Atlanta, Georgia, USA" (or "Atlanta, GA, USA")

- I don't think there are any references to the Grounded Theory method used. Can you please add a reference to the particular approach you followed?

- The reference to NVivo seems off. It can be referenced in the reference list without an URL within the text.

Ruwan Ratnayake, LSHTM